# Nonvolatile Ternary Memristor Based on Fluorene-Benzimidazole Copolymer/Au NP Composites

**DOI:** 10.3390/nano12234117

**Published:** 2022-11-22

**Authors:** Meng Gao, Yanting Du, Haifeng Yu, Zhaohua He, Shuhong Wang, Cheng Wang

**Affiliations:** 1School of Chemical Engineering and Materials, Heilongjiang University, Harbin 150080, China; 2Jieyang Branch of Chemistry and Chemical Engineering Guangdong Laboratory, Jieyang 515200, China

**Keywords:** Au NPs, donor–acceptor type, fluorene-containing copolymer, ternary storage

## Abstract

A donor–acceptor type polymer of poly [2,7-(9,9-dioctyl)-fluorene-*alt*-7H-benzimidazo-[2,1-a]benzo[de]isoquinolin-7-one] (PF-BBO) based on benzimidazole groups was synthesized. This material was incorporated into data storage devices that exhibited good data storage characteristics. In order to improve the storage properties of the device, Au NPs were compounded in this material. We observed an increase in the ratio of switching current for the device with the PF-BBO/Au NP composite as the active layer. The device comprising 8 wt% Au NPs demonstrated optimal storage performance with a switching current ratio of 1:3.4 × 10^2^:1.0 × 10^5^ and a threshold voltage of −0.40 V/−0.85 V, respectively. The number of cycle times of this device was over 3000, which indicates excellent stability. Thus, the devices containing PF-BBO/Au NP composite as active materials offer a new dimension for future application prospects of high-density data storage.

## 1. Introduction

An information-driven twenty-first century has significantly shifted our lifestyles and, as a result, increased the demand for more and more effective information storage capabilities, including storage of daily-collected data [1,2]. At the same time, modern electronic devices continue to decrease in size with compromising device integration and function. Eventually, traditional semiconductor devices will be unable to satisfy the rising demand for memory storage. As a result, new information storage solutions and technologies with drastically enhanced storage capacity and reading speed, as well as lower working voltage, will be soon required [3].

Recent research trends demonstrate that RRAM (resistive random-access memory) with the simple electrode/active layer/electrode structure [4] shows excellent data storage performance [5,6] for nanoscale size, high storage density, fast wipe speed, low operating voltage, and excellent cycle stability, among others [7,8]. These devices typically used inorganic materials as active materials, such as oxide [9,10,11,12], carbides [13,14,15], nitrides [16,17,18], and so on. Inorganic materials, however, have reached their physical limits. Some organic materials have gradually become a research focus, and are capable of significantly improving memristor performance [19]. Additionally, such organic materials can be easily processed and applied to substrates (e.g., by spin-coating) to fabricate memory storage devices [20,21,22,23]. Thus, memristor-based organic active layer materials offer great potential as they are easier to process, less expensive, and able to provide advanced properties for memory storage with a simple multilayered structure. All of these features make organic materials ideal candidates as active layers in resistive switching devices [24].

Organics are frequently referred to as organic bistable materials [19,25,26,27]. In general, the resistance switching mechanism varies with the organic thin-film material structure and kind [28]. For example, poly (N-vinyl carbazole) (PVK) [29,30] and polyfluorene derivatives (PFs) [21,31] can act as electron donors and hole migrators, assisting in their charge transfer and acting as charge capture sites. Storage devices with such matrices offer reproducible bistability [15,32]. However, these storage devices are mainly limited to binary storage characteristics [33]. Thus, for increasing storage capacity, multivariate data storage is needed, which is the focus of our research efforts. Some researchers demonstrated that storage devices featuring donor–acceptor (D–A) type conjugated polymer containing benzothiadiazole (BT) possessed outstanding ternary resistance switching performance, a recognizable current ratio, and a low threshold voltage [34]. Another promising device with ternary nonvolatile memory devices contained conjugated polymers with Ir (III) side-chain complexes [35]. All of the above research about the storage performance of organic thin film materials provides us with an inspiring idea for the design of data storage material. The research found that the most recent developments in the devices relevant to data storage involve organic/inorganic nanoparticle composite materials [36,37,38,39]. The nanoparticles in the active layer serve as trapping centers, facilitating charge injection and trapping. Thus, we anticipate that the introduction of metal nanoparticles into polymers will improve their data storage performance. Because of their large specific surface area and free electronic spatial restriction capability, Au NPs can convert energy levels [40]. Additionally, a relative simplicity of Au NPs’ synthesis makes it ideal for RRAM preparation. Therefore, we choose the Au NP composite with polymer for use as the active material to prepare the data storage device.

This paper reported a D–A type polymer of poly [2,7-(9,9-dioctyl)-fluorene-*alt*-7H-benzimidazo-[2,1-a]benzo[de]isoquinolin-7-one] (PF-BBO), which was synthesized by the Suzuki reaction. We prepared a series of composites varying PF-BBO/Au NP ratios and used them as active materials in data storage devices. The storage mechanism, stability, and memory performance of the devices were investigated. Our data indicated that a memristor containing the prepared polymer with 8 wt% of Au NPs possessed the best ternary storage performance.

## 2. Materials and Methods

### 2.1. Material

All of the chemicals and reagents were of analytical quality and were purchased from J&K Scientific and used as received. Detailed preparation procedures for the monomer (M-BBO), polymer (PF-BBO), and Au NPs are provided in the Supporting Information. These synthesis steps for both M-BBO and PF-BBO are outlined in Figure 1. The Au NPs were prepared by the sodium citrate reduction method and stored at a low temperature and vacuum environment. The Au NPs were subjected to X-ray diffraction (XRD) analysis. Peaks at 38.24°, 44.43°, 64.57°, and 77.45°, which correspond to the (111), (200), (220), and (311) crystal planes of Au, respectively, were observed (Figure 1a). Transmission electron microscopy (TEM) of the PF-BBO/Au NP composite thin films revealed that 8 wt% Au NPs were distributed in the PF-BBO randomly and uniformly (Figure 1b).

### 2.2. Storage Device Preparation Procedure

Indium tin oxide conductive glass (ITO), used as a substrate, was soaked in absolute ethanol before use. Several Au NP suspensions were prepared, during which we paid special attention to the temperature to avoid thermally induced Au NPs’ aggregation. These suspensions were then mixed with 3 mg/mL PF-BBO solution in toluene and ultrasonicated for 2 h. The mixed suspension was spin-coated on ITO substrate and dried in vacuum. After that, aluminum electrodes were deposited on top using the physical vapor deposition method to create an ITO/PF-BBO/Au/Al organic memory device. A device containing pure PF-BBO was also constructed. Figure 2 depicts a memory device schematic.

## 3. Results and Discussion

### 3.1. Characterization Test of PF-BBO

The molecular structure was characterized by NMR spectroscopy and infrared absorption spectroscopy, and the data characterization results were consistent with the expected results. Therefore, PF-BBO was confirmed to be successfully synthesized (shown in Appendix A).

The molecular weight of PF-BBO was measured by gel permeation chromatography (Appendix A). According to the results, PF-BBO possessed a molecular weight of 29,586 and a narrow polydispersity index of 1.03. Thermogravimetric analysis (TGA) of PF-BBO recorded in the N_2_ atmosphere showed a 10% weight loss at 412 °C (Figure 3), which is indicative of the excellent thermal stability of the PF-BBO. Hence, it can be applied as active material in storage devices at a higher temperature.

Furthermore, the optoelectronic performance of the PF-BBO was analyzed. The optoelectronic performance analysis characterizations of the PF-BBO were used to compute the HOMO and LUMO energy level, respectively. It can be shown in the Supporting Information (Appendix A). The results show that the conduction process of PF-BBO devices is dominated by hole injection.

### 3.2. SEM Images of the Device Structure

The three-layer structure of the device can be seen more conveniently in the SEM images. Cross-section SEM images of devices based on PF-BBO and PF-BBO/Au NP composites showed that the active material layer thicknesses were equal to 258.6 nm and 278.9 nm, respectively (shown in Figure 4a,b).

### 3.3. Storage Performance of the Device

Generally, conductivity and storage capacity can be seen in the I–V curves of device. As shown in Figure 5a, there were four stages in the I–V curve of the ITO/PF-BBO/Al device (device A, which did not include any Au NPs).

When the voltage was decreased from 0 V to −6 V (shown as red curve in Figure 5a), two current surges were observed at −1.70 V and −2.15 V. The following mechanism explains these two surges. Devices can switch between the low and high resistance states by switching between high (HRS, OFF), intermediate (IRS, ON1), and low states of resistivity (LRS, ON2). The “Write” event in the memory cell corresponds to this conduction process. During the second sweep (from −6 V to 0 V, the blue curve in Figure 5a), the ITO/PF-BBO/Al device remained stable and was in the LRS. As a result, this procedure was labeled as “Read”. Figure 5a shows that the third sweep (from 0 V to 6 V, shown in green color in Figure 5a) showed a rapid reduction in current at 3.9 V, following which the system reverted to the original HRS. This process was interpreted as an “Erase” event. During the final scan (from 6 V to 0 V, the black curve in Figure 5a), the device returns to its original state, which was assigned as “Rewrite” [41]. The two-stage current changes (e.g., sudden spike and drop) are typically referred to as “SET” and “RESET”, respectively [42,43]. Thus, the ITO/PF-BBO/Al device demonstrated rewritable electrical characteristics, which could also be interpreted as the flash-type ternary nonvolatile storage behavior. Throughout the entire I–V cycle, the threshold voltage *V*_th1_/*V*_th2_ and the switching current ratio were equal to −1.70 V/−2.15 V and 1:1.0 × 10^1^:1.6 × 10^2^, respectively.

Stability testing of the device is necessary. After the retention and durability tests, the test results are shown in Figure 5b,c. In the ITO/PF-BBO/Al device (device A), current retention tests at −1.00 V provided proof of their long-term stability. Even after 10^4^ s (over 3500 cycles), the current of device A remained constant, which indicates satisfactory stability.

In order to explore the influence of Au NPs’ content on the storage performance of the device, I–V curves of devices constructed with varying Au NPs’ concentrations were thus recorded (shown in Figure 6a). The ITO/PF-BBO/Au/Al devices labeled (B–F) contained 2, 4, 6, 8, and 9 wt% of Au NPs, respectively. Taking device B as an example, the change from HRS to LRS was clearly visible at −0.89 V. The threshold voltage *V*_th1_/*V*_th2_ for devices B, C, D, E, and F was equal to −0.89 V/−1.40 V, −0.60 V/−1.15 V, −0.49 V/−0.90 V, −0.40 V/−0.85 V, and −0.50 V/−0.90 V, respectively, while the corresponding switching current ratio for these devices was equal to 1:1.1 × 10^2^:1.6 × 10^4^, 1:5.8 × 10^2^:1.2 × 10^4^, 1:9.3 × 10^1^:2.2 × 10^4^, 1:3.4 × 10^2^:1.0 × 10^5^, and 1:5.3 × 10^1^:2.3 × 10^4^, respectively. Device E with 8 wt% of Au NPs had the largest switching current ratio and the lowest threshold voltage, which we considered to be the optimal performance. In addition, the retention and durability of device E exhibited optimal performance too. For device E, current retention tests at −1.00 V confirmed their long-term stability (shown in Figure 6b,c). Even after 10^4^ s (over 3500 cycles), the current of device E remained constant without significant fluctuations, which indicates outstanding stability and excellent long-term nonvolatile storage.

In order to effectively demonstrate how the recombination content of Au NPs affects the storage properties of the device, 30 groups of devices with different Au NPs’ content were prepared and tested. Twenty data points were chosen at random from the devices of each group and totaled in the histogram in Figure 7. The threshold voltage’s standard deviation and the switching current ratio were both very small. The graphs clearly demonstrate that the amount of Au NPs in the system obviously affects the switching current ratio and the threshold voltage. As the concentration of Au NPs in the ITO/PF-BBO/Au/Al device increases, the switching current ratio increases to its maximum at 8 wt% of Au NPs, then it starts to decrease. This can be explained by the fact that Au NPs possess electrical conductivity and, when dispersed in the thin film, electron transport passageways are possibly formed [44]. There will be an increase in the number of electron transport pathways as the concentration of Au NPs increases. The result shows that the switching current ratio of the Au-containing device significantly improved while the threshold voltage decreased dramatically. In order to compare the effect of Au NPs clearly, we plotted the voltage–switching current ratio curve of devices A and E, as shown in Figure 8. Compared with device A, the switching current ratio of device E (8 wt% Au NPs’ content) increases by approximately three orders of magnitude, reaching the maximum value. A high-conductivity condition develops in the composite layer, possibly as a result of the appropriate concentration of Au NPs, which lowers the switching current ratio.

Plotting an I–V curve on a logarithmic scale (for both x- and y-axes) can provide further details on the device switching mechanism as the plot sections can be fitted linearly (as shown in Figure 9a). At low voltages, the slope of the fitting is almost 1. The ratio between the current and voltage depends on ohmic conduction (OC), governed by the thermal excitation of current during the trap filling [45]. To some extent, semiconductors with local gap states can capture free charges during their transport, acting as electric charge traps. Trap-filling process-generated free carriers play a critical role in charge transfer at extremely low electric fields. As the intensity of the applied electric field increases, the carriers progressively fill the traps generated by structural defects and the Fermi level rises to the conductive band, termed the trap filling limited current (TFLC) zone. A fast increase in conductivity occurs when the PF-BBO film traps fill up at *V*_th1_. The conduction behavior at this stage is consistent with Child’s law, which states that the current is directly proportional to the square of the voltage. Therefore, space charge limited current (SCLC) dominates this stage. When the bias voltage exceeds *V*_th2_ and the carriers conduct electricity, the electric field becomes more intense and the device is forced into LRS.

The current in the low-voltage region of the log (I) versus log (V) curve recorded from the ITO/PF-BBO/Au/Al device exhibited ohmic conductivity with a slope equal to 1.11 (shown in Figure 9b). As bias voltage is increased, traps caused by structural defects become gradually filled, rising to the Fermi level [46]. The corresponding slope equals 1.80. At V = *V*_th1_, the conductivity rapidly rises as a result of the SCLC [47]. During this process, charge carriers fill all traps in the PF-BBO film. The accumulation of additional charges in the PF-BBO film reduces the bandgap [48]. It is worth noting that carriers migrate along the channel with the least resistance, generating the conductive routes of charge-trapping [49]. More conductive carriers appear on the ITO side and accumulate in the Al electrode when the voltage is raised further. At V = *V*_th2_, enough conductive carriers accumulate to connect the upper and lower electrodes, which converts the device conductivity into LRS. In this state, devices have minimal resistance until the reverse voltage is introduced. Consequently, the carriers are then successfully transmitted through the pathway and conductive filament [50]. As the voltage continues to ascend, an excessive number of carriers generate heat, which fractures the conductive filament [51,52]. In this case, the change happens from the low-conductivity state to the high-conductivity state in the device.

We have also studied PF-BBO utilizing molecular simulations and theoretical calculations based on density functional theory (DFT). The HOMO cloud exists predominantly on the fluorene unit, whereas the LUMO cloud exists predominantly on the M-BBO unit as the fluorene unit is a better electron donor than the M-BBO. Figure 10a shows the energy level diagram of the ITO/PF-BBO/Al device, which represents the storage procedure of the device. In comparison with the ITO electrode, the potential barrier between the Al electrode and the LUMO energy level is substantially larger. As a result, it is simpler to inject electrons into HOMO holes from the anode. These devices’ conductivity is regulated by a hole injection mechanism. Upon reaching the threshold voltage, the aggregated energy assists in carrier injection into the PF-BBO film, which results in a sharp current increase. This device corresponds to the transition from the OFF to the ON state. As the electric field is reversed, the captured charge carriers are separated from the PF-BBO film. Consequently, the storage device has no ability to remain in the ON state and switches back to the OFF state.

Figure 10b shows the DFT-calculated equivalent molecule surface electrostatic potential (ESP) of PF-BBO [41]. Regions marked with blue and white in the figure demonstrate a series of positive ESPs on the conjugated skeleton, which may serve as charge carrier transport channels. At the same time, the red area marks negative ESPs, formed on electron-deficient groups. These areas could be referred to as “charge traps” as they impede the charge carrier transport. Because the barrier is created by low voltage, carriers cannot readily travel through, keeping the device in the OFF state. Filling the charge trap occurs at the threshold voltage *V*_th1_ when charge carriers gather energy and transition from the donor to acceptor levels. This event corresponds to the transition from the OFF to the ON1 state. Charge carriers collect enough energy to fill the charge traps and attain an ON2 trap-free state as the voltage rises, resulting from the intensification of the electric field. After that, the captured charge carriers gradually transition to a state of charge separation [53]. The resulting charge is typically in a metastable state [54]. When the excess electric field is withdrawn or reversed, the conduction pathway is disconnected and, as a result, the storage device is unable to remain in the ON state [55]. Therefore, the device switches back to the HRS.

The rationale for the addition of Au NPs to the PF-BBO film was to improve its conductance and act as carrier-trapping centers to facilitate charge transport and successfully catch the charge carriers (shown in Figure 11) [56,57]. The process in the sweep 1 region is depicted in Figure 11a, which shows that, when the applied sweep voltage is low, the carriers are emitted from the ITO bottom electrode and injected into the PF-BBO matrix under the action of the electric field, but the carriers have not been captured by the Au NPs. When a voltage is provided to the sweep 2 region, as depicted in Figure 11b, the injected carriers gradually increase and are trapped by the trapping sites and interfaces in the Au NPs. When the applied scan voltage in sweep 3 is increased to an adequately high level, the injected carriers increase sharply and are almost completely trapped in the inter-layer (Figure 11c). On the contrary (sweep 4), when a negative scan voltage is applied to the device, the carriers trapped in the trapping sites will be released into the PF-BBO medium and then transferred to the bottom electrode of the ITO electrode, thereby causing the device to transition from an ON state to an OFF state (Figure 11d). The addition of Au NPs enhances charge injection and trapping during the charge transfer, which indicated that Au NPs acted as the carrier-capturing center. Moreover, the addition of Au NPs also increases the switching current ratio and reduces the threshold voltage of the device, which increases the misread rate and reduces energy consumption. At the same time, the devices still maintain the ternary memory characteristics.

## 4. Conclusions

The M-BBO monomer and the D–A type PF-BBO polymer were successfully synthesized and then used to construct electrically bistable ternary resistance memristors, which were prepared without and with Au NPs incorporated into the polymer. A device containing this D–A type PF-BBO/Au NP composite was extensively tested for storage performance. The maximum switching current ratio of the corresponding device increased from 1:1.0 × 10^1^:1.6 × 10^2^ to 1:3.4 × 10^2^:1.0 × 10^5^ and the threshold voltage decreased from −1.70 V/−2.15 V to −0.40 V/−0.85 V when the system was incorporated with 8 wt% of Au NPs, which indicates a significant improvement in the device storage performance. Even after 3 h of continuous testing (which corresponds to 3000 switching cycles), the device remained stable. The storage process of the PF-BBO/Au NP-based device was also assessed using the conductive filament mechanism. The addition of Au NPs enhances charge injection and trapping during the charge transfer, which indicated that Au NPs acted as the carrier-capturing center. Additionally, devices based on PF-BBO/Au NP composite material demonstrated a very high level of memory performance, substantially reduced energy consumption, and good stability. The approaches used in this work open new horizons in rationally designing multiple-level high-performance memory devices.

## Data Availability

Not applicable.

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
