# Peer review of "Nonvolatile Ternary Memristor Based on Fluorene-Benzimidazole Copolymer/Au NP Composites"

_nanomaterials, 2022, doi:10.3390/nano12234117_

Round 1

Reviewer 1 Report

The authors have presented a good quality and interesting results on Nonvolatile ternary memristor titled, “Nonvolatile ternary memristor based on fluorine-benzimidazole copolymer: Au NPs composites”

However, the reviewer suggests the authors to make minor revision to this version of the manuscript considering the below remarks/comments and suggestions so that the quality of the manuscript can be improved further. The suggested changes may be incorporated in the revised version of the manuscript.

Comments

·      *   Could you add clear details on the formation of Au NC with respect to the size of the NC (diameter) (approximate value) in this work and Au NC Size dependence on the annealing temperature?

·       *  In this work, you reported that the thicknesses of active material layer are 258.6 and 278.9nm. why did you choose these specific thicknesses for this memristor performance? How does the performance of the memristor affect with variation in the active layer thickness?

·      *   What is the range of active layer thickness where the memristor shows acceptable performance?

·      *      Will you see the similar trend as shown in Fig. 5 for different thicknesses?

·     *    In Fig. 6(a), consistency of the intermediate (IRS, ON1) state is not very clear. What parameters affect the formation of intermediate (IRS, ON1) state?

·    *   In Fig. 7(b), is shows threshold voltage variation with respect Au NPs content. In this manuscript authors used 8wt%. What is the threshold voltage variation across a number of devices with 8wt% Au NPs content?

·       *  Is ON/OFF current ratio (high value) the only parameter you considered for choosing 8wt% Au NPs content?

*     Is this memristor device appropriate for mass production? Is this device compatible with Silicon technology for integration?

·      *     Check for typos and grammatical errors.

Reviewer 2 Report

The authors have reported that the nonvolatile ternary memristor is based on fluorine-benzimidazole copolymer: Au NPs composites. This manuscript is not clear and needs improvement in presentation. To publish the journal, the manuscript can be accepted after revising the following errors.

1. The major defect of this study is the debate or argument is not clearly stated in the introduction session. Hence, the contribution is weak in this manuscript. I suggest the author enhance your theoretical discussion and arrive at your debate or argument.

2. In Section result and discussion, authors should improve the result interpretation. All graphs are not systematic. Please pay more attention to the section result and discussion. This section should be systematized.

3. The author should replace the electrical storage word with data storage.

4. The author has mentioned that Au NPs can improve the photoelectric properties but there are no results related to this statement. The author should perform related experiments or remove the relevant sentences.

5. There are several typographical issues. An author should carefully revise it for such errors.

Round 2

Reviewer 2 Report

I think it was improved along comments, hence I recommend it can be published on Nanomaterials.